# Context-Aware Object Detection With Convolutional Neural Networks

## Abstract

Although the state-of-the-art object detection methods are successful in detecting and classifying objects by leveraging deep convolutional neural networks (CNNs), these methods overlook the semantic context which implies the probabilities that different classes of objects occur jointly. In this work, we propose a context-aware CNN (or conCNN for short) that for the first time effectively enforces the semantics context constraints in the CNN-based object detector by leveraging the popular conditional random field (CRF) model in CNN. In particular, conCNN features a context-aware module that naturally models the mean-field inference method for CRF using a stack of common CNN operations. It can be seamlessly plugged into any existing region-based object detection paradigm. Our experiments using COCO datasets showcase that conCNN improves the average precision (AP) of object detection by 2 percentage points, while only introducing negligible extra training overheads.

## 1 Introduction

In recent years, deep convolutional neural networks (CNN) (Goodfellow et al., 2016) have been used with great success in object detection tasks. However, some well known principles in computer vision that have been shown to be effective in object detection are largely overlooked. In particular, the vision community has shown that the semantic context, namely the correlations among the objects, helps object detection Rabinovich et al. (2007). That is, when performing the task of object detection, objects' class labels should be inferred with respect to other objects in the scene. For example, when detecting objects in a scene of a baseball match in the COCO dataset, which contains four objects: "Baseball", "Baseball bat", "human", and "baseball glove", the CNN-based object detection model tends to recognize the "baseball bat" as "tooth brush". Yet it can easily be seen that this object of "tooth brush" does not fit into the context with other labels. This is because the state-of-the-art object detection methods (He et al., 2015; Girshick, 2015; Ren et al., 2015; Dai et al., 2016; Lin et al., 2017; He et al., 2017) adopt the region-based paradigm since it was introduced in the seminal R-CNN work (Girshick et al., 2014). Given a set of region proposals, these methods perform object classification and bounding box regression on each proposal *individually*, without taking the semantic context into consideration.

In this work, we now propose a context-aware object detection strategy called conCNN to address the above shortcoming. By seamlessly plugging a *context-aware module* into the existing CNN-based object detection paradigm, conCNN automatically learns and effectively enforces the semantics context constraints, yet requiring minimal modification of the existing deep object detection architecture.

conCNN is inspired by our observation that probabilistic graphical models, in particular, Conditional Random Fields (CRFs) have been successful in enhancing the accuracy of low level computer vision tasks (Ladicky et al., 2009; Zheng et al., 2015; Rabinovich et al., 2007; Krähenbühl & Koltun, 2011) such as image segmentation or object detection. More specifically, CRF models the label assignment problem as a probabilistic inference problem such that constraints can be incorporated, for example, the label agreement between similar pixels in image segmentation.

We thus design a context-aware module that for the first time leverages CRF in deep object detection architecture to maximize the contextual agreement among the objects in the same scene, yet only slightly increasing the training time of the object detection model. The overall context-aware deep object detection network, corresponds to a deep architecture with a traditional CNN-based module concatenated by a context-aware module, can be trained end-to-end following the common practice of back-propagation in deep neural networks. Therefore, we successfully combine the strengths of both CNN and CRF-based graphical models in one unified framework.

More specifically, the context-aware module reformulates the *iterative mean-field* inference (Krähenbühl & Koltun, 2011) – a popular approximate inference method for CRF as a stack of common CNN layers. This way, it learns a compatibility matrix as normal parameters of deep neural network that represents the probabilities of label co-occurrences, as the compatibility function in CRF.

Our experimental evaluation on the COCO datasets confirms that conCNN improves the AP of object detection by 2 percentage points.

## 2 Network Architecture of conCNN

In this work, we proposal a context-aware neural network (conCNN) for object detection. Essentially, we achieve this by plugging a context-aware module into a CNN network built upon the popular region-based object detection paradigm as shown in Fig. 1.

**Region-based Object Detection.** The region-based object detection paradigm (Girshick et al., 2014) first processes the whole image with several convolutional and max pooling layers (the CNN backbone) to produce a conv feature map. Taking the feature map as input, the Region Proposal Network (RPN in short) generates a set of rectangular object proposals that most likely contain objects. Then for each object proposal, a region of interest (RoI) pooling layer extracts a fixed-length feature vector from the feature map.

Each feature vector is fed into a sequence of fully connected layers (FC layers) that finally branch into two sibling output layers corresponding to the classification task and the bounding box regression task. The classification task produces softmax probability estimates over $L$ object classes. The bounding box regression task outputs four real-valued numbers for each of the $L$ object classes. Each set of 4 values encodes refined bounding box positions for one of the $L$ classes.

**Context-aware Module.** Our context-aware module is plugged into the classification task. Specifically, it takes the prediction from the last FC layer in the classification task as well as the location for each bounding box in the regression task as input and produces a modified prediction that incorporates the context semantics. The modified prediction distribution is then fed into the softmax layer to generate the final probability estimates.

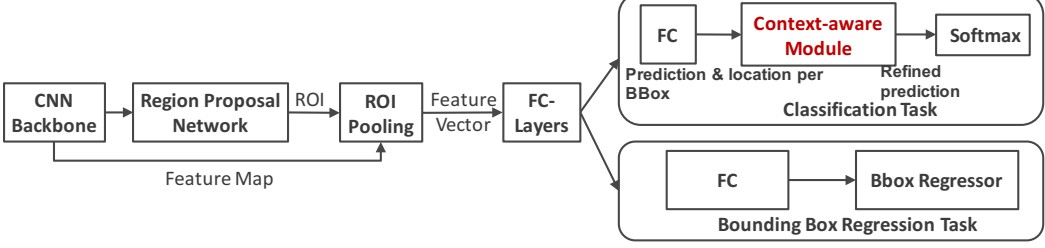

Figure 1: Overall Network Architecture of conCNN

## 3 Conditional Random Field in Object Detection

In this section we briefly overview Conditional Random Fields (CRF) for object detection. In object detection, a CRF models object labels (classes) as random variables that form a Markov Random Field (MRF) when conditioned upon the input image.

**Conditional Random Field.** Let $X_i$ be the random variable w.r.t. object $i$. It represents the label assigned to the object $i$ and can take any value from a pre-defined set of labels $\mathcal{L} = \{l_1, l_2, ..., l_L\}$.

Let $\mathbf{X}$ be the vector formed by the random variables $X_1$, $X_2$, ..., $X_N$, where $N$ denotes the number of objects in the image $I$. Then the probability of some label assignment $x$ conditional on the global observation $I$ is given by the CRF model:

$$P(X = x|I) = \frac{1}{Z(I)} exp(-E(x)) \tag{1}$$

Here $E(x)$ denotes the energy of the configuration $x \in \mathcal{L}^N$, while $Z(x)$ denotes the partition function (Lafferty et al., 2001).

### 3.1 Mean-field Approximation

Minimizing the above CRF energy E(x) yields the most probable label assignment $x$ (maximizing $P(x)$) for the given image. Since this exact minimization is intractable, in the fully connected pairwise CRF model, a mean-field approximation to the CRF distribution (Krähenbühl & Koltun, 2011) is used for approximate maximum posterior marginal inference. It approximates the CRF distribution $P(\mathbf{X})$ using a simpler distribution $Q(\mathbf{X})$, which can be expressed as the product of independent marginal distributions, $Q(\mathbf{X}) = \prod_i^N Q_i(X_i)$, where $Q_i(X_i)$ corresponds to one object $i$ in image $I$. Leveraging Krähenbühl & Koltun (2011) for object detection, we use the following update equation to iteratively compute a valid $Q_i(X_i)$:

$$Q_i(X_i = l) = \frac{1}{Z(I)} exp \left\{ \underbrace{-\psi_u(X_i = l)}_{Unary\ Component} - \underbrace{\sum_{l' \in \mathcal{L}} \mu(l, l') \sum_{m=1}^{K} w^m \sum_{i \neq j} k^m(f_i, f_j) Q_j(X_j = l')}_{Pairwise\ Component} \right\} \tag{2}$$

In Equation 2, the unary component $-\psi_u(X_i = l)$ (denoted as $U_i(l)$) measures the probability that the label $l$ is assigned to the object $i$ ($X_i = l$). As shown in Fig. 1, in our conCNN it is obtained from the classification task, which assigns label to object $i$ without considering the context semantics.

The pairwise component $\sum_{l' \in \mathcal{L}} \mu(l, l') \sum_{m=1}^{K} w^m \sum_{i \neq j} k^m(f_i, f_j) Q_j(x_j = l')$ reflects the pairwise influence of the label assignment ($X_j = l'$) of objects $j$ on the label assignment of object $i$ ($X_i = l$). Generally speaking, The pairwise component measures the cost of assigning labels $l$, $l'$ to objects $i$, $j$ simultaneously. It enforces the context semantics constraints.

Among this pairwise component, function $\mu(l, l')$ captures the compatibility (or the possibility of co-occurrence) between a pair of labels $l$ and $l'$. It corresponds to the new parameters that have to be learned in the training of conCNN to enforce the context semantics constraints.

Each function $k^m(f_i, f_j)$ for m = 1, ..., M, represents a function applied on feature vectors ($f_i$, $f_j$) of objects $i$ and $j$. The feature vector $f_i$ of object $i$ corresponds to some characteristics of object $i$ derived from a CNN. In our scenarios, $f_i$ corresponds to the location of the bounding box or the probability estimation of object $i$ taking label $l \in \mathcal{L}$. Essentially, $k^m(f_i, f_j)$ reflects how strongly an object $i$ is related to other objects $j$, hence called *relation function*.

Note in Krähenbühl & Koltun (2011) and Zheng et al. (2015) which apply CRF in image segmentation, $k^m(f_i, f_j)$ corresponds to the Gaussian kernel. However, in object detection Gaussian kernel does not necessarily work well (Rabinovich et al., 2007). In Sec. 4.2, we will introduce in more details the relation functions used in our work.

---

**Algorithm 1** Mean-field for dense pairwise CRFs

---

1: $Q_i(l) \leftarrow \frac{1}{Z(I)} exp(U_i(l));$           ▷ Initialization

2: **while** not converged **do**

3:     $\hat{Q}_i^m(l) \leftarrow \sum_{j \neq i} k^m(f_i, f_j) Q_j(l);$           ▷ Message passing from all $X_j$ to all $X_i$

4:     $\hat{Q}_i(l) \leftarrow \sum_m w^m \hat{Q}_i^m(l);$           ▷ Weighting

5:     $\hat{Q}_i(l) \leftarrow \sum_{l' \in \mathcal{L}} \mu(l, l') \hat{Q}_i(l');$           ▷ Compatibility transform

6:     $\hat{Q}_i(l) \leftarrow U_i(l) - \hat{Q}_i(l);$           ▷ Local Update

7:     $Q_i(l) \leftarrow \frac{1}{Z(I)} exp(\hat{Q}_i(l));$           ▷ Normalization

8: **end while**

---

## 3.2 Mean-field Algorithm

The update equation (Equation 2) leads to the mean-field algorithm summarized in Algorithm 1 (Krähenbühl & Koltun, 2011). Each iteration of Algorithm 1 performs a message passing step, a compatibility transform, a local update, and a normalization.

Next, we show how to reformulate all these steps in Algorithm 1 as common CNN layers and hence effectively enforces the context semantics constraints in CNN-based object detection, using the classical practice of back propagation in deep neural networks.

## 4 CRF Inference As Common CNN Layers

To reformulate the mean-field inference as common CNN layers, it is essential to be able to calculate error differentials in each layer w.r.t. its inputs in order to back-propagate the error differentials to previous layers during training. This way, CRF parameters such as the label compatibility function $\mu(l, l')$ can be learned automatically during the training of the full network. Therefore, in this section we also discuss how to calculate error differentials with respect to the parameters in each layer.

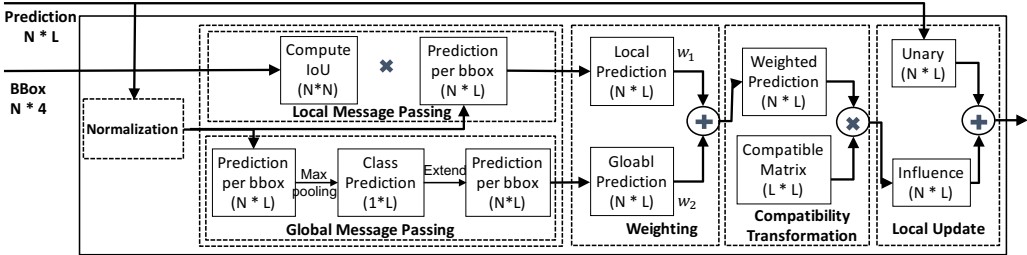

Figure 2: Context-aware Module. L stands for the number of object classes and N stands for the number of objects in image I.

## 4.1 Initialization

In this step (Line 1, Algorithm 1), the initialization operation $Q_i(l) = \frac{1}{Z(I)} exp(U_i(l))$ is performed. Since $Z(I)$ (Lafferty et al., 2001) corresponds to $\sum_{l \in \mathcal{L}} exp(U_i(l))$, we get:

$$Q_i(l) = \frac{1}{Z(I)} exp(U_i(l)) = \frac{exp(U_i(l))}{\sum_{l \in \mathcal{L}} exp(U_i(l))} \tag{3}$$

Note that the form of $Q_i(l)$ in Equation 3 is equivalent to the softmax function (Goodfellow et al., 2016), widely used in CNN architectures. Naturally it supports back-propagation. That is, during back-propagation its error differentials received at the output of this step is passed back to the input of the context-aware module (the prediction produced by bounding box classification) after performing backward pass calculations of the softmax function.

## 4.2 MESSAGE PASSING

The message passing step updates each single variable $X_i$ w.r.t. object $i$ by aggregating information from all other variables $X_j$ w.r.t. other objects $j$. As shown in Fig. 2, conCNN supports two types of information aggregation, called local message passing and global message passing.

**Local Message Passing.** Local message passing models the geometry relationship among the objects in the same image. Similar to Hu et al. (2018), it is based on the intuition that given an object $i$, the objects $j$ tend to have strong influence on $i$ if its bounding box overlaps with the bounding box of $i$. Therefore, the local message passing takes the location information into consideration when aggregating the influence from other objects.

More specifically, we first compute the Intersect of Union (IoU) between the bounding boxes of a pair of objects. The IoUs w.r.t. all object pairs correspond to a $N \times N$ matrix, where $N$ denotes the number of objects in image $I$. To compute the IoU matrix, we need as input the bounding box information of all $N$ objects which can be considered as a $N \times 4$ matrix. To efficiently compute the IoU matrix, in this work we formulate its computation as a matrix transformation operation such that this process can be performed on GPU in parallel. The diagonals of the IoU matrix are then reset to zeros because we only need to aggregate the predictions from other objects.

Once the IoU matrix is ready, the local message passing process can be mapped to a matrix product operation between the $N \times N$ IoU matrix and a $N \times L$ prediction matrix which correspond to the original predictions w.r.t. the $N$ objects and the $L$ classes. This results in the final aggregated predictions (a $N \times L$ matrix), weighting the predictions from the overlapping bounding boxes more.

Since this step is equivalent to a set of linear weighted-sum transformations and does not involve any parameters, the back-propagation is trivially supported.

**Global Message Passing.** Inherent co-occurrence relations also exist among the objects in the same scene, not tied to the geometry locations of these objects. For instance, although keyboard and mouse do not overlap, the two object classes together define a computer-related context and can help differentiate mouse and soap which look similar in appearance.

In this work, this o-occurrence relationship is captured by the global message passing. It uses the probability that each object class $l$ appears in an image to model the context of this image, denoted as $p(l)$. Here we define $p(l)$ as the maximal probability estimation of class $l$ among all objects in the image, that is:

$$p(l) = max\{p_i(l)|\forall object\ i \in I\} \tag{4}$$

where $p_i(l)$ denotes the probability that the classification task assigns object $i$ to class $l$.

The $p(l)$ w.r.t. all $l \in \mathcal{L}$ can be computed by applying a global max pooling function (Goodfellow et al., 2016) on the $N \times L$ prediction matrix. This produces a $1 \times L$ class probability vector, which as the global context of this image, influences all objects equally.

Since max pooling is a common operation in CNN, the back-propagation can be naturally supported in the typical way.

## 4.3 WEIGHTING STEP

The weighting step takes a weighted sum of the $M$ outputs of the message passing step, corresponding to the $M$ relation functions. This step can be mapped to a $1 \times 1$ convolution filter with $M$ input channels and 1 output channel. Here each input channel corresponds to a $N \times L$ matrix. The $1 \times 1 \times M \times 1$ parameters of this filter correspond to the weights $w^m$ of the $M$ relation functions. As a common convolution operation, error derivative can be computed in the usual manner to pass the error derivatives back to the previous step.

### 4.4 Compatibility Transform

The outputs $\hat{Q}_i(l)$ on different labels $l \in \mathcal{L}$ influence each other to a variant extent, depending on the compatibility between these labels. In conCNN, $\forall$ two labels $l$ and $l' \in \mathcal{L}$, the compatibility is parameterized by the compatibility function $\mu(l, l')$, corresponding to a $L \times L$ compatibility matrix $\mathbb{C}$.

Organizing $\hat{Q}_i(l)$ with respect to all $N$ objects in image $I$ and all labels $l \in \mathcal{L}$ as a $N \times L$ matrix $\hat{\mathbb{Q}}$, the compatibility transform step can be implemented as a matrix product: $\hat{\mathbb{Q}} \times \mathbb{C}$. The output is also a $N \times L$ matrix.

In CNN, this matrix product can be viewed as a common convolution filter, where the spatial receptive field of the filter is $1 \times 1$, and the number of input and output channels are both $L$. The $1 \times 1 \times L \times L$ weights of this filter correspond to the compatibility matrix $\mathbb{C}$. In other words, learning the label compatibility function $\mu$ is equivalent to learning the weights of this filter. Since this step is a common convolution operation, transferring error differentials from the output to its input can be done in the usual way.

### 4.5 Local Update

In this step (Line 6, Algorithm 1), the output $\hat{Q}_i(l)$ from the compatibility transform step is subtracted from the unary input $U_i(l)$. Since this step does not have any parameters, transferring error differentials can be done trivially by copying the differentials at the output of this step to both inputs $\hat{Q}_i(l)$ and $U_i(l)$.

### 4.6 Normalization

Similar to the initialization step, the normalization step (Line 7, Algorithm 1) corresponds to a softmax operation without parameters.

## 5 Experimental Evaluation

### 5.1 Overview of Experimental Setting

**Datasets.** We demonstrate the effectiveness of our context-aware module using the benchmark COCO dataset and a subset of it.

The original COCO dataset contains 80 categories. As in previous work (He et al., 2017; Lin et al., 2017), we train using the union of 80k train images and a 35k subset of validation images (trainval35k), and evaluate on the remaining 5k validation images (minival).

The COCO subset dataset contains 6 categories of the original COCO including baseball bat, baseball glove, sink, toilet, mouse and keyboard. These 6 categories represent 3 different scenes (outdoor, bathroom, indoor) with 2 classes from each scene. There are 13k training images and 531 validation images in this COCO subset.

**Methodology.** We evaluate: (1) Faster R-CNN (Lin et al., 2017) as baseline; (2) Relation Network: Faster R-CNN with Relation Module (Hu et al., 2018); (3) Our method conCNN: Faster R-CNN with our context-aware module (Sec. 2). We ran experiments on 4 P100 GPU instances on Google cloud. The results show that our conCNN outperforms Relation Network in improving the AP of object recognition.

**Evaluation Metrics.** We report the standard COCO metrics including AP (averaged over IoU thresholds from 0.5 to 0.95), AP50 (IoU = 0.5), AP75 (IoU = 0.75) and AP at different scales including $AP_S$ (Small), $AP_M$ (Medium), $AP_L$ (Large).

### 5.2 Implementation Details

All approaches use ResNet-101-FPN as backbone. We implemented our conCNN by plugging the context-aware module into the penultimate layer of the classification task. We set

hyper-parameters following the existing Faster R-CNN work (Lin et al., 2017; Ren et al., 2015). The evaluation of the baseline (Lin et al., 2017) and Relation Network (Hu et al., 2018) is based on the models published by the authors of Relation Network.

During training, we adopt image-centric training (Girshick, 2015). Images are resized such that their scale (shorter edge) is 800 pixels as done in (Lin et al., 2017). An RoI is considered positive if it has IoU with a ground-truth box of at least 0.5 and negative otherwise. Each mini-batch has 4 images per GPU and each image has 512 sampled RoIs, with a ratio of 1:3 of positives to negatives. For the COCO subset with 6 categories, we train on 4 GPUs for 12k iterations, with a learning rate of 0.02 which is decreased by 10 at the 9kth iteration. For the COCO dataset with 80 categories, we train on 4 GPUs for 90k iterations, with a learning rate of 0.02 which is decreased by 10 at the 60kth and 80kth iteration. At test time, the proposal number is 1000 (as in Lin et al. (2017)).

## 5.3 EXPERIMENTS ON COCO SUBSET WITH 6 CATEGORIES

We first test our conCNN on the COCO subset with 6 categories. This subset contains images from 3 different scenes including outdoor, bathroom, and indoor. We use this subset to validate that our conCNN can effectively capture the co-occurrence relationship between the objects in the same scene.

Table 1: Results on COCO (6 categories).

| Methods | Backbone | AP | $AP_{50}$ | $AP_{75}$ | $AP_S$ | $AP_M$ | $AP_L$ |
|---|---|---|---|---|---|---|---|
| Faster R-CNN | Resnet-101-FPN | 41.6 | 67.7 | 44.5 | 21.1 | 47.1 | 51.2 |
| Faster RCNN + Relation | Resnet-101-FPN | 40.6 | 66.8 | 44.1 | 23.1 | 45.4 | 48.9 |
| conCNN | Resnet-101-FPN | 44.07 | 70.57 | 48.34 | 24.54 | 48.47 | 44.94 |

As shown in Table 1, in almost all cases, our conCNN outperforms Faster R-CNN and Relation Network. The performance gain results from the context-aware module in our conCNN, which leverages both the geometry and co-occurrence relationships among the objects. In particular, conCNN is good at detecting small and medium objects. Although small and medium objects are difficult to recognize purely by appearance, the relationships among the objects can help emphasize the correlated objects in the same image and thus improve the $AP_S$ and $AP_M$. Relation network does not perform well on this dataset because the Relation Network only considers the geometry relationship among the objects overlapping with each other, while objects hardly overlap in this dataset.

In addition, our conCNN introduces negligible training overhead. Specifically, the training takes 4.12 hours and 4.24 hours for Faster R-CNN and conCNN respectively. Relation Network takes 5.02 hours for training, because it uses a more complex relation module.

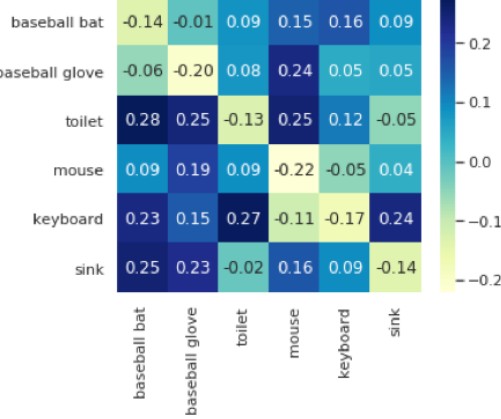

Figure 3: Compatibility Matrix for COCO with 6 Categories.

Fig. 3 shows the compatibility matrix learnt by conCNN. By Eq. 2, correlated classes should have negative values in this matrix. The smaller the value is, the stronger the correlation between the pair of classes. As expected, the learnt compatibility matrix shows that baseball glove is strongly correlated to baseball bat, while toilet/mouse is highly correlated to sink/keyboard. This confirms the effectiveness of conCNN in capturing context semantics.

## 5.4 Experiments on COCO with 80 categories

We also test our conCNN on the whole COCO dataset with 80 categories. As shown in Table 2, our conCNN still outperforms Faster R-CNN and Relation Network. Relation network performs well when detecting large objects. That is as expected because the Relation Network only considers the geometry relationship among the objects overlapping with each other, while large objects tend to overlap with more objects than small objects. The training time of conCNN is 27.25 hours – only 0.75 hours longer than the Faster R-CNN which takes 26.5 hours. On the other hand, the Relation Network takes 32.25 hours to train.

Table 2: Results on COCO (80 categories).

| Methods | Backbone | AP | $AP_{50}$ | $AP_{75}$ | $AP_S$ | $AP_M$ | $AP_L$ |
|---|---|---|---|---|---|---|---|
| Faster R-CNN | Resnet-101-FPN | 36.6 | 59.3 | 39.3 | 20.3 | 40.5 | 49.4 |
| Faster RCNN + Relation | Resnet-101-FPN | 38.6 | 59.9 | 43 | 22.1 | 42.3 | 52.8 |
| conCNN | Resnet-101-FPN | 39.48 | 61.05 | 43.20 | 22.67 | 42.58 | 52 |

## 6 Related Work

**CRF in Computer Vision.** Zheng et al. (2015) combines CRF and CNN to perform pixel-level semantic *image segmentation*. It formulates the CRF as an RNN. The RNN is then plugged in as a part of a CNN which can be trained end-to-end using back-propagation. Our work instead focuses on leveraging CRF in *object detection*. Instead of modeling the training process of CRF as a RNN, we factorize the mean-field approximate inference of CRF as common CNN layers and use the common CNN operations such as softmax and convolution to naturally simulate CRF in enforcing the context semantics constraints. Further, unlike Zheng et al. (2015) which uses Gaussian kernels to encourage label agreement between similar pixels, we design customized functions for object detection task. These functions effectively reflect the co-occurrence relationship among the objects in the same scene. CRF was also used as post-processing to refine the pixel-level label predictions or object-level class assignment (Rabinovich et al., 2007; Ladicky et al., 2009; Krähenbühl & Koltun, 2011). Since CRF is not incorporated into the training process of segmentation or object detection model, these methods are shown to be not as effective as Zheng et al. (2015).

**Object Relation in Deep Learning.** Hu et al. (2018) proposed a relation network that adapts the concept of attention in natural language processing (NLP) into the classical CNN-based object detection framework. The key idea is to use attention to model the geometry relationships among the objects in the same image, such as *a plate* on top of *a table* or *a tooth brush* in *a cup*. These dependencies are represented as relation features which are used together with other features to classify objects. Our work instead combines the strengths of CNN and CRF. It not only models the geometry relationships among the objects in one image, but also leverages the inherent co-occurrence relationships between object classes, such as mouse and keyboard, fridge and microwave, or toilet and sink.

**Object Relation in Classical Computation Vision.** Similar to the use of CRF, most object detection works before deep learning arises use object relations as a post-processing step (Divvala et al., 2009; Galleguillos et al., 2008; Mottaghi et al., 2014; Torralba et al., 2003; Tu & Bai, 2010; Rabinovich et al., 2007; Felzenszwalb et al., 2010). The detected objects are re-scored by considering object relationship such as the co-occurrence of different object classes (Rabinovich et al., 2007; Felzenszwalb et al., 2010). However, unlike our work, in this setup the object detector is unaware of the object relations during the training phase. Therefore, it cannot fully employ the object relations in object detection.

## 7 Conclusion

In this work, we proposed a context-aware neural network conCNN that takes the context semantics into consideration in object detection. The key idea is to embed a context-aware module into the CNN-based object detection network that effectively simulates the learning process of Conditional Random Field (CRF) model using a stack of common CNN operations. Combining the strengths of both CNN and CRF, conCNN effectively improve the AP of object detection as confirmed in our experiments on COCO datasets.

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
