# OpenReview forum: "Context-Aware Object Detection With Convolutional Neural Networks"
_ICLR.cc/2020/Conference — Reject_

### Official Review · AnonReviewer3 · 2019-10-20
**Official Blind Review #3**

**Rating:** 3

**Review:**

This paper introduces a context-aware neural network (conCNN) that integrates context semantics into account for object detection. The proposed approach achieves this by embedding a context-aware module into the Faster R-CNN detection framework. The context-aware module simulates the learning process of Conditional Random Fields (CRF) model using a stack of common CNN operations. Specifically, this paper employs the mean-field approach of [1]. Experiments are performed on COCO dataset.

The  paper reads well. The idea of combining the strengths of both CNNs and CRF-based graphical models in a detection framework is interesting. Though the main idea is borrowed from the mean-field approach of [1], it has been successfully reformulated as a stack of common CNN layers. However, my main concern is that the experimental results (Page 8 Table 2) does not support the merits of the proposed approach. When comparing with the existing Faster R-CNN + Relation work of [2], the improvement provided by the proposed approach is negligible. In fact the proposed approach achieves inferior results on large objects (APL), compared to [2]. [1] uses attention to model the geometric relationships among objects in the same image. In addition to geometric relationships, the proposed approach leverages the inherent co-occurence relationships between object classes. However, this additional information does not seem to provide much help by looking at the results on COCO dataset. The paper does present an experiment to justify this additional information in Table 1. However, that small experiment is only on 6 object categories and does not seem to generalize when going towards full COCO dataset with 80 categories (Table 2). Therefore, the reviewer would recommend to thoroughly evaluate the merits of the proposed approach in depth on COCO and on additional datasets (i.e., Visual Genome).

[1] Philipp Krähenbühl, Vladlen Koltun: Efficient Inference in Fully Connected CRFs with Gaussian Edge Potentials. NIPS 2011.
[2] Han Hu, Jiayuan Gu, Zheng Zhang, Jifeng Dai, Yichen Wei: Relation Networks for Object Detection. CVPR 2018.


**Experience Assessment:**

I have published in this field for several years.

**Review Assessment: Checking Correctness Of Derivations And Theory:**

I assessed the sensibility of the derivations and theory.

**Review Assessment: Checking Correctness Of Experiments:**

I carefully checked the experiments.

**Review Assessment: Thoroughness In Paper Reading:**

I read the paper thoroughly.

---

> ### Author Response · Authors · 2019-11-15
> **Response to Reviewer 3: The merits of our method**
>
> Thank you for the comments. Here we respond to your main concern.
>
> Comment: "My main concern is that the experimental results (Page 8 Table 2) does not support the merits of the proposed approach. When comparing with the existing Faster R-CNN + Relation work of [2], the improvement provided by the proposed approach is negligible."
>
> Thank you for the comment. Here we respond to your main concern.
>
> Comment: "My main concern is that the experimental results (Page 8 Table 2) does not support the merits of the proposed approach. When comparing with the existing Faster R-CNN + Relation work of [2], the improvement provided by the proposed approach is negligible."
>
> Response: As shown in Table 2, our average precision of conCNN on all 80 COCO categories  is 39.48, while the average precision of Faster R-CNN + Relation (Relation Network) is 38.6. On this challenging COCO dataset, we believe the gain of our model is reasonable. Moreover, our model is much simpler than Relation Networks. As shown in our experiments, conCNN converges faster than Relation Network -- almost as fast as the original Faster R-CNN. The reason is that compared to Relation Networks, we introduce many fewer additional parameters to the original Faster R-CNN model to enforce the semantics constraints. In addition, based on our evaluation, our conCNN model is much more stable than Relation Networks, meaning conCNN always converges to an average precision better than that of Faster R-CNN, while we have to carefully tune Relation Networks to beat the Faster R-CNN baseline. In summary, conCNN is simpler yet more accurate and more stable than Faster R-CNN.

---

### Official Review · AnonReviewer1 · 2019-10-30
**Official Blind Review #1**

**Rating:** 3

**Review:**

The paper proposes a contextual reasoning module following the approach proposed by the NIPS 2011 paper for object detection. Specifically, the algorithm proposed by NIPS 2011 are first converted to end-to-end modules step-by-step, and then added to the detection framework Faster R-CNN. The comparison is only to one other baseline (Relation Module), and some improvements are show especially for small objects.

+ Context reasoning is an important but unsolved problem in vision, it is a good attempt to apply CRF to object detection

- The subset results are not convincing. For example, AP_L has decreased dramatically from 51.2 to 44.9. This is shocking. Large objects should maintain a similar performance if the contextual module really works. This result is actually very negative to me.
- I am not sure the idea of applying conCNN to 6 categories of COCO is a great idea to begin with: context reasoning requires a lot of categories, especially those with few examples to benchmark with. If the categories are supplied with a lot of training examples, it is very hard for context to really help there. I would at least stick to full set of COCO (I am fine with reducing the number of training images), or switch to the recently proposed LVIS dataset:
https://www.lvisdataset.org/
- I think the work did a less comprehensive literature survey for object detection with context reasoning. For example:
Qi, Lu, et al. "Sequential context encoding for duplicate removal." Advances in Neural Information Processing Systems. 2018.
Chen, Xinlei, and Abhinav Gupta. "Spatial memory for context reasoning in object detection." Proceedings of the IEEE International Conference on Computer Vision. 2017.
There are actually many other attempts from different groups that try to nail down this problem, please at least review them in the paper.

Overall this is a paper that's not ready. There might be technical contributions (still with the presence of Zheng et al ICCV 2015, it is limited), but the paper needs to be refined in many ways to get accepted.

**Experience Assessment:**

I have published one or two papers in this area.

**Review Assessment: Checking Correctness Of Derivations And Theory:**

I did not assess the derivations or theory.

**Review Assessment: Checking Correctness Of Experiments:**

I assessed the sensibility of the experiments.

**Review Assessment: Thoroughness In Paper Reading:**

I read the paper at least twice and used my best judgement in assessing the paper.

---

> ### Author Response · Authors · 2019-11-15
> **Response to Reviewer 1: Explanation on Lower AP_L**
>
> We thank the reviewer for the valuable comments. Here we summarize and respond to the .
>
> Comment 1: "The subset results are not convincing. For example, AP_L has decreased dramatically from 51.2 to 44.9. This is shocking. Large objects should maintain a similar performance if the contextual module really works. This result is actually very negative to me."
>
> Response: On the COCO subset with 6 classes, the AP_L of our approach is indeed lower than the original Faster R-CNN. We believe this is because of the specificity of the 6 classes we pick, since the Faster R-CNN + Relation method also has a AP_L Lower than Faster R-CNN. Furthermore, on the whole COCO dataset with 80 classes, both our model (conCNN) and Faster R-CNN + Relation have a higher AP_L than Faster R-CNN as shown in Table 2.
>
> Comment2: Running experiments on the recently proposed LVIS dataset instead of the 6 class COCO
>
> Response: Thanks for the good suggestion. We are running experiments on this challenging LVIS dataset.
>
> Comment3: The missed related work
>
> Response: Thank you for pointing us these works. We will cover these in our related work.

---

### Official Review · AnonReviewer2 · 2019-10-30
**Official Blind Review #2**

**Rating:** 3

**Review:**

This paper proposes a CRF-based context module for CNN-based object detectors. In particular for the two-stage region-based detector, like Faster RCNN, the context module is added right before the output layer of the classification head. Every box proposed by the RPN is a node in the CRF, and its label is the classification label. Message passing is unrolled as neural network layers. Potentials are defined based on object detector outputs, box overlap, and co-occurrence of class labels. Experiments are performed on the MS COCO object detection task.

The proposed method is similar to the CRF-based method for FCN segmentation networks except that the nodes are boxes instead of pixels. And simpler ideas had been explored in the pre-DL era. In this sense, the main idea of this work is incrementally novel.

The presentation of the paper is fine in general. But the “global message passing” paragraph in Sec 4.2 can be improved. In particular, how p(l) is used in message passing? It is better to provide more specific descriptions so that the paper is self-contained.

The proposed model outperformed the baseline with a reasonably significant margin. However, its average accuracy on all 80 COCO categories is no better than “the Faster RCNN + Relation”. Since the proposed model is also not significantly simpler than the Relation Network, the experimental results do not establish a new state-of-the-art.

The experimental results are also a bit thin. More ablative studies can be helpful, e.g., global message parsing/local message passing.

Given the moderate novelty (which is good but not good enough as a standalone reason to accept this paper) and the not-strong-enough experimental results, I feel more work is needed for the paper to be readily publishable.


**Experience Assessment:**

I have published one or two papers in this area.

**Review Assessment: Checking Correctness Of Derivations And Theory:**

I assessed the sensibility of the derivations and theory.

**Review Assessment: Checking Correctness Of Experiments:**

I assessed the sensibility of the experiments.

**Review Assessment: Thoroughness In Paper Reading:**

I made a quick assessment of this paper.

---

> ### Author Response · Authors · 2019-11-15
> **Response to Reviewer 2: The significance of our work**
>
> Thanks for the reviews. These are valuable to improve our work. Here is one review we want to clarify and we believe is important.
>
> Comment: "However, its average accuracy on all 80 COCO categories is no better than “the Faster R-CNN + Relation”. Since the proposed model is also not significantly simpler than the Relation Network, the experimental results do not establish a new state-of-the-art. "
>
> Response: First, as shown in Table 2, our average precision on all 80 COCO categories is indeed higher than Faster R-CNN + Relation (Relation Network). More specifically, the average precision of conCNN is 39.48, while the average precision of Faster R-CNN + Relation is 38.6. Moreover, our model is much simpler than Relation Networks. As shown in our experiments,  the training time of our model is 27.25 hours – only 0.75 hours longer than the Faster R-CNN which takes 26.5 hours, while the Relation Network takes 32.25 hours to train. The reason is that compared to Relation Network, we introduce many fewer additional parameters to the original Faster R-CNN model to enforce the semantics context constraints. In addition, based on experiments, our model is also more stable than Relation Networks -- we always converge to an average precision higher than that of Faster R-CNN, while we have to carefully tune Relation Networks to beat the Faster R-CNN baseline. In short, our model is simpler yet more accurate and stable than Faster R-CNN. So we do believe our work pushes the state-of-the-art forward.
>
> We are also running new experiments on the LVIS dataset per the suggestion of Reviewer 1. This dataset is challenging. It will still take us additional time to make sure that the comparison is fair and convincing before we can comfortably report the results.

---

### Decision · Program_Chairs · 2019-12-19

**Decision:**

Reject

**Comment:**

The paper proposes a contextual reasoning module following the approach proposed by the NIPS 2011 paper for object detection. Although the reviewers find the proposed approach reasonable, the experimental results are weak and noisy. Multiple reviewers believe that the paper will benefit from another review cycle, pointing out that the authors response confirmed that multiple additional (or redoing of) experiments are needed.